# DON'T REINVENT THE STEERING WHEEL

## ABSTRACT

To make safe and informed decisions, autonomous driving systems can benefit from the capability of predicting the intentions and trajectories of other agents on the road in real-time. Trajectory forecasting for traffic scenarios has seen great strides in recent years in parallel with advancements in attention-based network architectures and robust, large-scale benchmarks. However, such models are becoming larger, resource-hungry, and less portable as state-of-the-art pushes for larger-scale of road networks and real-world complexity. Previous works that achieve state-of-the-art results predict future trajectories as a series of waypoints in Euclidean space, yet do not frame learning through the lenses of classical kinematic models that describe the motion of moving vehicles. Instead of leaving the network to learn the inherent dynamics of traffic agents, we can instead leverage kinematic models of vehicle dynamics as *priors* to guide neural networks toward physics-informed solutions earlier in learning. By combining existing knowledge of how agents move with powerful deep learning techniques, agents learn trajectories that are not only more interpretable but also more plausible in terms of vehicle kinematic constraints. In this work, we investigate the use of different kinematic formulations as learning priors for trajectory forecasting tasks and evaluate how each affects learning both empirically and analytically. In addition, we take advantage of time integration in order to derive the original output format of future trajectory coordinates, enabling the use of existing architectures and complementing previous work. This approach is easy to implement for trajectory forecasting and achieves a considerable performance gain on large-scale benchmarks.

## 1 INTRODUCTION

Rapid improvements in deep learning research directly bolster improvements on learning-based tasks in autonomous driving. In the last five years, perception and planning models for autonomous driving have not only seen great progress but also scaled well in size. For example, NVIDIA's end-to-end driving network (Bojarski et al., 2016) in 2016, contained about 250 thousand parameters on a convolutional neural network (CNN) architecture, while vision transformer (ViT) based model TransFuser (Chitta et al., 2022) appeared in 2022, now has over 168 million parameters! Similar patterns have emerged for other autonomous driving tasks, such as trajectory forecasting and small-scale agent simulation, where model size and complexity continue to grow. As model sizes become bigger and performance increases, so does the complexity of learning the basics of vehicle dynamics, where most vehicles travel in relatively straight lines behind the vehicle directly in front.

Since training a deep neural network is costly, it would be beneficial on both resource consumption and technique generalization to incorporate the use of existing dynamics models in the training process. Kinematic models have been widely used in robotics and simulation research for solving classical robot constraint and path planning problems. These models explicitly describe how changes in the input parameters influence the output of the dynamical system. These input parameters are often provided by either the robot policy as an action, or by a human in direct interaction with the robot, e.g. steering, throttle, and brake for driving a vehicle. For tasks modeling decision-making, such as trajectory forecasting of traffic agents, modeling the input parameters may be more descriptive and interpretable than modeling the output directly. Moreover, kinematic models relate the input actions directly to the output observation; thus, any output of the kinematic model should, at the very least, be physically feasible in the real world.

We hypothesize that learning first- or second-order terms via a differentiable kinematic model for trajectory forecasting will lead to better generalization of existing methods in various settings, whether it is smaller models, smaller datasets, or in the presence of noise.

In this paper, we present a method for incorporating car-following models as priors into modern trajectory forecasting network architectures. By leveraging the advantages of both network complexity and simplicity of traffic equations, the network can prioritize learning the complex patterns of human behavior *on top* of basic vehicle dynamics, which can be directly modeled with kinematics.

In summary, the main contributions of this work include:

1. A simple and effective method for incorporating kinematic priors into probabilistic models for trajectory forecasting (Section 4), which boosts performance (Section 5);

2. Results and analysis on three different kinematic formulations: velocity components, acceleration components, and speed + heading components (Section 4.2).

3. Analytical error bounds for the first and second-order kinematic formulations (section 4.3).

## 2 RELATED WORKS

### 2.1 TRAJECTORY FORECASTING FOR TRAFFIC

Traffic trajectory forecasting is a popular task where the goal is to predict the short-term future trajectory of multiple agents in a traffic scene. Being able to predict the future positions and intents of each vehicle provide context for other modules in autonomous driving, such as path planning. Large, robust benchmarks such as the Waymo Motion Dataset (Ettinger et al., 2021; Chen et al., 2023), Argoverse (Wilson et al., 2021), and the NuScenes Dataset (Caesar et al., 2020) have provided a standardized setting for advancements in the task, with leaderboards showing clear rankings for state-of-the-art models. Amongst the top performing architectures, most are based on Transformers for feature extraction (Shi et al., 2022; Qian et al., 2023; Liu et al., 2021; Zhou et al., 2023; Ngiam et al., 2021; Girgis et al., 2021). Current SOTA models also model trajectory prediction probabilistically, as inspired by the use of GMMs in Multipath (Chai et al., 2020b).

One common theme amongst all state-of-the-art, however, is that none employ elements of classical kinematics in combination with powerful attention-based networks. In our work, we present a method for use of kinematic priors which can be complemented with any previous work in trajectory forecasting. Our contribution can be implemented in any of the SOTA methods above, since it is a simple reformulation of the task with no additional information needed.

### 2.2 PHYSICS-BASED PRIORS FOR LEARNING

Model-based learning has shown to be effective in many applications, especially in robotics and graphics. There are generally two approaches to using models of the real world: 1) learning a model of dynamics via a separate neural network (Deisenroth & Rasmussen, 2011; Rempe et al., 2022; Lutter et al., 2019; Greydanus et al., 2019; Janner et al., 2019), or 2) using existing models of the real world via differentiable simulation (de Avila Belbute-Peres et al., 2018; Degrave et al., 2019; Geilinger et al., 2020; Qiao et al., 2020; Freeman et al., 2021; Son et al., 2023; Xu et al., 2021; Liang et al., 2019).

In our method, we pursue the latter method. Since we are not modeling complex systems such as cloth or fluid, simulation of traffic agent states require only a simple, fast, and differentiable update. In addition, since the kinematic models do not describe interactions between agents, the complexity of the necessary model is greatly reduced. Many current SOTA trajectory prediction models are designed intentionally to study the relationships between agents via self-attention; thus, in our method, we leave the complex tasks to the network, and model the simple tasks (e.g., how a vehicle moves forward) to the kinematic model.

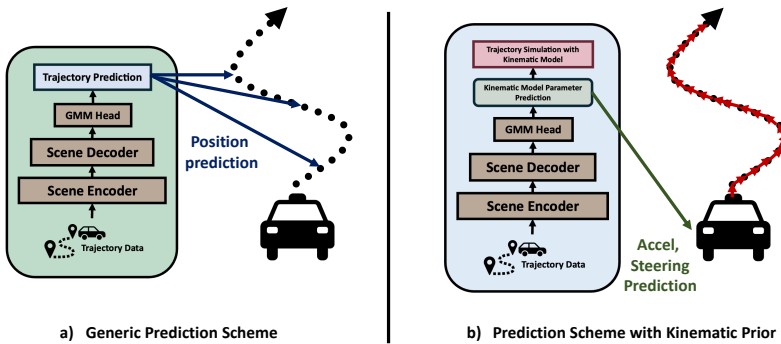

Figure 1: **Generic trajectory prediction versus prediction with a kinematic prior.** In generic prediction schemes (a), the model will directly predict trajectory positions across time. With a kinematic prior (b), the model predicts first or second-order terms instead of the positions directly, then obtain positions via time integration following the kinematic model.

## 3  KINEMATICS OF TRAFFIC AGENTS

Our method borrows concepts from simulation and kinematics. In a traffic simulation, each vehicle holds some sort of state consisting of position along the global x-axis $x$, position along the global y-axis $y$, velocity $v$, and heading $\theta$. This state is propagated forward in time via a kinematic model describing the constraints of movement with respect to some input acceleration $a$ and steering angle $\delta$ actions. The Bicycle Model, a classical and popularly utilized in path planning for robots, describes the kinematic dynamics of a wheeled agent given its length $L$:

$$\frac{d}{dt}\begin{pmatrix} x \\ y \\ \theta \\ v \end{pmatrix} = \begin{pmatrix} \dot{x} \\ \dot{y} \\ \dot{\theta} \\ \dot{v} \end{pmatrix} = \begin{pmatrix} v \cdot cos\theta \\ v \cdot sin\theta \\ \frac{v \cdot tan\theta}{L} \\ a \end{pmatrix}$$

We refer to this model throughout our method to derive the relationship between predicted distributions of kinematic variables and the corresponding distributions of positions $x$ and $y$ for the objective trajectory prediction task.

We use Euler time integration in forward simulation of the kinematic model to obtain future positions. As we will show later, explicit Euler time integration is simple for handling Gaussian distributions, despite being less precise than higher-order methods like Runge-Kutta. In other words, an agent's state is propagated forward from timestep $t$ to $t + 1$ with the following, given a timestep interval $\Delta t$:

$$\begin{pmatrix} x_{t+1} \\ y_{t+1} \\ \theta_{t+1} \\ v_{t+1} \end{pmatrix} = \begin{pmatrix} x_t \\ y_t \\ \theta_t \\ v_t \end{pmatrix} + \begin{pmatrix} \dot{x} \\ \dot{y} \\ \dot{\theta} \\ \dot{v} \end{pmatrix} \cdot \Delta t$$

## 4  METHODOLOGY

In this section, we discuss in detail four different kinematic formulations as priors for current trajectory forecasting methods.

### 4.1  PROBABILISTIC TRAJECTORY FORECASTING

Trajectory forecasting is a popular task in autonomous systems where the objective is to predict the future trajectory of multiple agents for $T$ total future timesteps, given a short trajectory history. Re-

cently, state-of-the-art methods (Chai et al., 2020a; Varadarajan et al., 2022; Wang et al., 2023; Shi et al., 2022) utilize Gaussian Mixture Models (GMMs) to model the distribution of potential future trajectories, given some intention waypoint or destination of the agent and various extracted agent or map features. Each method utilizes GMMs slightly differently, however, all methods use GMMs to model the distribution of future agent trajectories. We apply a kinematic prior to the GMM head directly, thus our method is agnostic to the design of the learning framework. Instead of predicting a future trajectory deterministically, current works instead predict a mixture of Gaussian components $(\mu_x, \mu_y, \sigma_x, \sigma_y, \rho)$ describing the mean $\mu$ and standard deviation $\sigma$ of $x$ and $y$, in addition to a correlation coefficient $\rho$ and Gaussian component probability $p$. The standard deviation terms, $\sigma_x$ and $\sigma_y$, along with correlation coefficient $\rho$, parameterize the covariance matrix of a Gaussian centered around $\mu_x$ and $\mu_y$.

Ultimately, the prediction objective is, for each timestep, to maximize the log-likelihood of the ground truth trajectory waypoint $(x, y)$ belonging to the position distribution outputted by the GMM:

$$\mathcal{L} = -\log p_h - \log \mathcal{N}_h(x - \hat{\mu}_x, \hat{\sigma}_x; y - \hat{\mu}_y; \rho)$$

This formulation assumes that distributions between timesteps are conditionally independent, similar to (Chai et al., 2020a) and its derivatives. Alternatively, it's possible to implement predictions with GMMs in an autoregressive manner, where trajectory distributions are dependent on the position of the previous timestep. The drawback of this is the additional overhead of computing conditional distributions with recurrent architectures, rather than jointly predicting for all timesteps at once. In the following section, we derive the relationship between each timestep distribution through several kinematic formulations, then use Euler time integration to apply transformations to distributions at all timesteps. In other words, we use conditionally independent kinematic parameter distributions to derive conditionally dependent position distributions.

### 4.2 KINEMATIC PRIORS FOR GAUSSIAN MIXTURE MODEL PREDICTIONS

The high-level idea for kinematic priors is simple: instead of predicting the distribution of positions at each timestep, we can instead predict the distribution of first-order or second-order kinematic terms and then use time integration to derive the subsequent position distributions.

The intuition for enforcing kinematic priors comes from the idea that even conditionally independent predicted trajectory waypoints have inherent relationships with each other depending on the state of the agent, even if the neural network does not model it. By propagating these relationships across the time horizon, we focus optimization of the network in the space of kinematically feasible trajectories. We consider three different formulations: 1) with velocity components $v_x$ and $v_y$, 2) with acceleration components $a_x$ and $a_y$, and finally 3) with speed $s = \|\boldsymbol{v}\|$ and heading $\theta$.

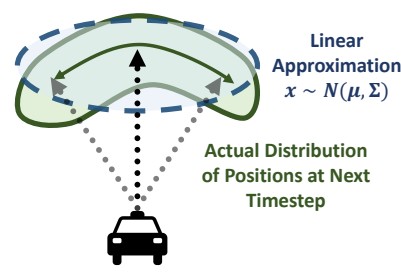

Figure 2: Linear approximation of position distributions via kinematic prior.

Additionally, we derive the analytical form with direction-less acceleration $a = \|\boldsymbol{a}\|$ and steering angle $\delta$ in Section A.1 of the Appendix.

#### 4.2.1 FORMULATION 1: VELOCITY COMPONENTS

The velocity component formulation is the simplest kinematic formulation, where the GMMs predict the distribution of velocity components $v_x$ and $v_y$ for each timestep $t$.

Our goal is to derive $(\mu_x^{t+1}, \mu_y^{t+1}, \sigma_x^{t+1}, \sigma_y^{t+1})$ given $(\mu_{v_x}^t, \mu_{v_y}^t, \sigma_{v_x}^t, \sigma_{v_y}^t)$.

In the deterministic setting, the position at the next timestep can be generated via Euler time integration given a time interval (in seconds) $\Delta t$, which varies depending on the dataset:

$$x^{t+1} = x^t + v_x^t \cdot \Delta t$$

If we consider both $x$ and $v$ to be Gaussian distributions rather than scalar values, we can represent the above in terms of distribution parameters below with the reparameterization trick used in Variational Autoencoders (VAEs) (Kingma & Welling, 2022):

$$\mathcal{N}_{x^{t+1}} = (\mu_x^t + \sigma_x^t \cdot \epsilon_x) + (\mu_{v_x}^t + \sigma_{v_x}^t \cdot \epsilon_v) \cdot \Delta t$$

where $\epsilon_x, \epsilon_v \sim \mathcal{N}(0, 1)$. By grouping deterministic (without $\epsilon$) and perturbed terms (with $\epsilon$), we obtain the reparameterized form of the Gaussian distribution describing $x^{t+1}$:

$$\mathcal{N}_{x^{t+1}} = (\mu_x^t + \mu_{v_x}^t \cdot \Delta t) + (\sigma_x^t \cdot \epsilon_x + \sigma_{v_x}^t \cdot \Delta t \cdot \epsilon_v)$$
$$\mathcal{N}_{y^{t+1}} = (\mu_y^t + \mu_{v_y}^t \cdot \Delta t) + (\sigma_y^t \cdot \epsilon_y + \sigma_{v_y}^t \cdot \Delta t \cdot \epsilon_v)$$
$$\mu_x^{t+1} = \mu_x^t + \mu_{v_x}^t \cdot \Delta t, \; \sigma_x^{t+1} = \sigma_x^t + \sigma_{v_x}^t \cdot \Delta t$$
$$\mu_y^{t+1} = \mu_y^t + \mu_{v_y}^t \cdot \Delta t, \; \sigma_y^{t+1} = \sigma_y^t + \sigma_{v_y}^t \cdot \Delta t$$

Also, for the first prediction timestep, we consider the starting trajectory position to represent a distribution with standard deviation equal to zero.

This Gaussian form is also intuitive as the sum of two Gaussian random variables is also Gaussian. Since this formulation is not dependent on any term outside of timestep $t$, distributions for all $T$ timesteps can be computed with vectorized cumulative sum operations. We derive the same distributions in the following sections in a similar fashion with different kinematic parameterizations.

**Linear interpolation based on timestep.** In addition to the three formulations described above, we also show results for Formulation 1 (velocity components) with linear interpolation of trajectories based on timestep. Although the Waymo dataset samples trajectories at 10hz, or 10 frames per second, which is extraordinarily granular for real world measurements, this sampling rate may also produce accumulating errors when used in simulation problems ($\Delta t = 0.1$). We implement an alternative version of Formulation 1 where predicted velocities are interpolated to smaller "sub-timesteps" where $\Delta t = 0.01$.

### 4.2.2 FORMULATION 2: ACCELERATION COMPONENTS

Following directly from the first formulation above, we now consider the second-order case where the GMM predicts acceleration components. Here, our goal is to derive $(\mu_x^{t+1}, \mu_y^{t+1}, \sigma_x^{t+1}, \sigma_y^{t+1})$ given $(\mu_{a_x}^t, \mu_{a_y}^t, \sigma_{a_x}^t, \sigma_{a_y}^t)$. The deterministic relationship between acceleration components $a_x$ and $a_y$ with $v_x$ and $v_y$ via Euler time integration is simply

$$v_x^{t+1} = v_x^t + a_x^t \cdot \Delta t$$
$$v_y^{t+1} = v_y^t + a_y^t \cdot \Delta t$$

Following similar steps to Formulation 1, we obtain the parameterized distributions of $v_x$ and $v_y$:

$$\mu_{v_x}^{t+1} = \mu_{v_x}^t + \mu_{a_x}^t \cdot \Delta t, \; \sigma_{v_x}^{t+1} = \sigma_{v_x}^t + \sigma_{a_x}^t \cdot \Delta t$$
$$\mu_{v_y}^{t+1} = \mu_{v_y}^t + \mu_{a_y}^t \cdot \Delta t, \; \sigma_{v_y}^{t+1} = \sigma_{v_y}^t + \sigma_{a_y}^t \cdot \Delta t$$

When computed for all timesteps, we now have $T$ total distributions representing $v_x$ and $v_y$, which then degenerates to Formulation 1 in Section 4.2.1.

### 4.2.3 FORMULATION 3: SPEED AND HEADING

Now, we derive the approximated analytical form of position distributions according to first-order dynamics of the Bicycle Model, speed $s = \|\boldsymbol{v}\|$ and heading $\theta$. Similarly as before, our goal is to derive $(\mu_x^{t+1}, \mu_y^{t+1}, \sigma_x^{t+1}, \sigma_y^{t+1})$ given $(\mu_s^t, \mu_\theta^t, \sigma_s^t, \sigma_\theta^t)$. Deterministically, we can get the update for $x^{t+1}$ and $y^{t+1}$ with the following relation from the Bicycle Model:

$$\begin{bmatrix} x^{t+1} \\ y^{t+1} \end{bmatrix} = \begin{bmatrix} x^t + s^t \cdot \cos\theta \cdot \Delta t \\ y^t + s^t \cdot \sin\theta \cdot \Delta t \end{bmatrix}$$

When representing this formulation in terms of Gaussian parameters, we notice that the functions $cos(\cdot)$ and $sin(\cdot)$ applied on Gaussian random variables do not produce Gaussians:

$$\begin{bmatrix} \mathcal{N}_{x^{t+1}} \\ \mathcal{N}_{y^{t+1}} \end{bmatrix} = \begin{bmatrix} (\mu_x^t + \sigma_s^t \cdot \epsilon_x) + (\mu_s^t + \sigma_s^t \cdot \epsilon_s) \cdot \cos(\mu_\theta^t + \sigma_\theta^t \cdot \epsilon_x) \cdot \Delta t \\ (\mu_y^t + \sigma_s^t \cdot \epsilon_y) + (\mu_s^t + \sigma_s^t \cdot \epsilon_s) \cdot \sin(\mu_\theta^t + \sigma_\theta^t \cdot \epsilon_y) \cdot \Delta t \end{bmatrix}$$

To amend this, we instead replace $cos(\cdot)$ and $sin(\cdot)$ with linear approximations $T(\cdot)$ evaluated at $\mu_\theta$.

$$T_{\sin}(\theta) = \sin(\mu_\theta) + \cos(\mu_\theta) \cdot (\theta - \mu_\theta)$$
$$T_{\cos}(\theta) = \cos(\mu_\theta) - \sin(\mu_\theta) \cdot (\theta - \mu_\theta)$$

We now derive the formulation of the distribution of positions with the linear approximations instead:

$$\begin{bmatrix} \mathcal{N}_{x^{t+1}} \\ \mathcal{N}_{y^{t+1}} \end{bmatrix} = \begin{bmatrix} (\mu_x^t + \sigma_x^t \cdot \epsilon_x) + (\mu_s^t + \sigma_s^t \cdot \epsilon_s) \cdot T_{\cos}(\mu_\theta^t + \sigma_\theta^t \cdot \epsilon_x) \cdot \Delta t \\ (\mu_y^t + \sigma_y^t \cdot \epsilon_y) + (\mu_s^t + \sigma_s^t \cdot \epsilon_s) \cdot T_{\sin}(\mu_\theta^t + \sigma_\theta^t \cdot \epsilon_y) \cdot \Delta t \end{bmatrix}$$

$$= \begin{bmatrix} (\mu_x^t + \sigma_x^t \cdot \epsilon_x) + (\cos(\mu_\theta) - \sin(\mu_\theta) \cdot \sigma_\theta \cdot \epsilon_\theta) \cdot (\mu_s + \sigma_s \cdot \epsilon_s) \cdot \Delta t \\ (\mu_y^t + \sigma_y^t \cdot \epsilon_y) + (\sin(\mu_\theta) + \cos(\mu_\theta) \cdot \sigma_\theta \cdot \epsilon_\theta) \cdot (\mu_s + \sigma_s \cdot \epsilon_s) \cdot \Delta t \end{bmatrix}$$

$$= \begin{bmatrix} \left( \begin{array}{c} \mu_x^t + (\mu_s \cdot \cos(\mu_\theta) \cdot \Delta t) \\ +\sigma_x^t \cdot \epsilon_x - (\mu_s \cdot \sigma_\theta \cdot \sin(\mu_\theta) \cdot \Delta t) \cdot \epsilon_\theta + (\sigma_s \cdot \cos(\mu_\theta) \cdot \Delta t) \cdot \epsilon_s \\ -(\sigma_s \cdot \sigma_\theta \cdot \sin(\mu_\theta) \cdot \Delta t) \cdot \epsilon_s \cdot \epsilon_\theta \end{array} \right) \\ \left( \begin{array}{c} \mu_y^t + (\mu_s \cdot \sin(\mu_\theta) \cdot \Delta t) \\ +\sigma_y^t \cdot \epsilon_y + (\mu_s \cdot \sigma_\theta \cdot \cos(\mu_\theta) \cdot \Delta t) \cdot \epsilon_\theta + (\sigma_s \cdot \sin(\mu_\theta) \cdot \Delta t) \cdot \epsilon_s \\ +(\sigma_s \cdot \sigma_\theta \cdot \cos(\mu_\theta) \cdot \Delta t) \cdot \epsilon_s \cdot \epsilon_\theta \end{array} \right) \end{bmatrix}$$

Notice that the last term of both entries involve the product of two univariate Gaussian random variables. Since both $\epsilon_s, \epsilon_\theta \sim \mathcal{N}(0, 1)$, we know that the product density of $\epsilon_s \epsilon_\theta$ produces an unnormalized Gaussian PDF with mean 0 and variance $\frac{1}{\sqrt{2}}$, per the proof from (Bromiley, 2003). Thus, we treat the last terms with $\epsilon_s \epsilon_\theta$ as belonging to a third scaled standard normal distribution, $\frac{1}{\sqrt{2}} \epsilon_\gamma$. The resulting approximated distributions of position following this becomes, approximately,

$$\mu_x^{t+1} = \mu_x^t + \mu_s \cdot \cos(\mu_\theta) \cdot \Delta t$$
$$\sigma_x^{t+1} = \sigma_x^t - \mu_s \cdot \sigma_\theta \cdot \sin(\mu_\theta) \cdot \Delta t + \sigma_s \cdot \cos(\mu_\theta) \cdot \Delta t - \frac{1}{\sqrt{2}}(\sigma_s \cdot \sigma_\theta \cdot \sin(\mu_\theta) \cdot \Delta t)$$
$$\mu_y^{t+1} = \mu_y^t + \mu_s \cdot \sin(\mu_\theta) \cdot \Delta t$$
$$\sigma_y^{t+1} = \sigma_y^t + \mu_s \cdot \sigma_\theta \cdot \cos(\mu_\theta) \cdot \Delta t + \sigma_s \cdot \sin(\mu_\theta) \cdot \Delta t + \frac{1}{\sqrt{2}}(\sigma_s \cdot \sigma_\theta \cdot \cos(\mu_\theta) \cdot \Delta t)$$

### 4.3 ERROR BOUND OF LINEAR APPROXIMATION.

We analytically derive the error bound for the linear approximation for $f(x) = \cos(x)$ and $f(x) = \sin(x)$ functions at $\mu_\theta$. Since the Taylor series expansion of both functions are alternating, the error is bounded by the term representing the second order derivative:

$$R_2^{\cos}(\mu_\theta + \sigma_\theta \cdot \epsilon_\theta) \leq \left| \frac{-\cos(\mu_\theta)}{2!} \right| (\mu_\theta + \sigma_\theta \cdot \epsilon_\theta - \mu_\theta)^2 = \left| \frac{-\cos(\mu_\theta)}{2} \right| \sigma_\theta^2 \cdot \epsilon_\theta^2 \leq \frac{\sigma_\theta^2}{2} \cdot \epsilon_\theta^2$$

$$R_2^{\sin}(\mu_\theta + \sigma_\theta \cdot \epsilon_\theta) \leq \left| \frac{-\sin(\mu_\theta)}{2!} \right| (\mu_\theta + \sigma_\theta \cdot \epsilon_\theta - \mu_\theta)^2 = \left| \frac{-\sin(\mu_\theta)}{2} \right| \sigma_\theta^2 \cdot \epsilon_\theta^2 \leq \frac{\sigma_\theta^2}{2} \cdot \epsilon_\theta^2$$

For both functions, the Lagrange error $R(x) = f(x) - T(x)$ is bounded by $R(x) \leq \frac{\sigma_\theta^2}{2} \cdot \epsilon_\theta^2 \leq \frac{\sigma_\theta^2}{2}$.

## 5 RESULTS

In this section, we show experiments which highlight the effect of kinematic priors on performance. We implement kinematic priors on state-of-the-art method Motion Transformer (MTR) Shi et al. (2022), which serves as our baseline method.

We train all experiments on eight RTX A5000 GPUs, with 64 GB of memory and 32 CPU cores. Experiments on the full dataset are trained for 30 epochs, while experiments with the smaller dataset are trained for 50 epochs. Additionally, we downscale the model from its original size of 65 million parameters to 2 million parameters and reproduce baseline results in our experiments. More details on training hyperparameters can be found in Table 8 of the appendix.

## 5.1 Performance on Waymo Motion Prediction Dataset

We evaluate the baseline model and all kinematic formulations on the Waymo Motion Prediction Dataset (Ettinger et al., 2021). The Waymo dataset consists of over 100,000 segments of traffic, where each scenario contains multiple agents of three classes: vehicles, pedestrians, and cyclists. The data is collected from high-quality, high-resolution sensors which sample traffic states at 10hz. The objective is, given 1 second of trajectory history for each vehicle, predict trajectories for the next 8 seconds. For simplicity, we use the bicycle kinematic model for all three classes and leave discerning between the three, especially for pedestrians, to future work.

We evaluate our model's performance on Mean Average Precision (mAP), Minimum Average Displacement Error (minADE), minimum final displacement error (minFDE), and Miss Rate, similarly to (Ettinger et al., 2021). We reiterate their definitions below for convenience.

- Mean Average Precision (mAP): mAP is computed across all classes of trajectories. The classes include straight, straight-left, straight-right, left, right, left u-turn, right u-turn, and stationary. For each prediction, one true positive is chosen based on the highest confidence trajectory within a defined threshold of the ground truth trajectory, while all other predictions are assigned a false positive. Intuitively, the mAP metric describes prediction precision while accounting for all trajectory class types. This is beneficial especially when there is an imbalance of classes in the dataset (e.g., there may be many more straight-line trajectories in the dataset than there are right u-turns).

- Minimum Average Displacement Error (minADE): The average L2 norm between the ground truth and the closest prediction; $minADE(G) = \min_i \frac{1}{T} \sum_{t=1}^{T} \|\hat{s}_G^t - s_G^{it}\|_2$.

- Minimum final displacement error (minFDE): The L2 norm between only the positions at the final timestep, $T$; $minFDE(G) = \min_i \|\hat{s}_G^T - s_G^{iT}\|_2$.

- Miss Rate: The number of predictions lying outside a reasonable threshold from the ground truth. The miss rate first describes the ratio of object predictions lying outside a threshold from the ground truth to the total number of agents predicted.

We show results for the Waymo Motion dataset in Table 1, where we compared the % difference in performance compared to the baseline for each formulation. From these results, we observe improvement over the baseline with Formulation 1, which involves the velocity components. Interestingly, there were less noticeable differences with other formulations when models were trained on the full dataset, though *Formulation 1 achieving a 2% increase in performance on the mAP metric.*

We also consider the effects of suboptimal settings for trajectory prediction, as we hypothesize that learning first or second-order terms provide information when data cannot. This is also motivated by problems in the real world, where sensors may not be as high-quality or specific traffic scenarios may not be so abundantly represented in data.

Table 1: **% difference in performance for vehicles on each kinematic formulation versus SOTA baseline on Waymo Motion Dataset, Marginal Trajectory Prediction.** We compare our model against the state-of-the-art architecture, Motion Transformer Shi et al. (2022). In our experiments, we downscale the backbone model size from **65 million** parameters to **2 million** parameters. From the results, we find that **Formulation 1 (velocity components) provides the greatest and most consistent boost in performance across most metrics over the baseline that does not include kinematic priors.**

| Method | ($\Delta\%$) mAP$\uparrow$ | ($\Delta\%$) minADE$\downarrow$ | ($\Delta\%$) minFDE$\downarrow$ | ($\Delta\%$) MissRate$\downarrow$ |
|---|---|---|---|---|
| Baseline | 0 | 0 | 0 | 0 |
| Ours + Formulation 1 | **2.376** | **-0.3444** | **-0.9102** | -0.3853 |
| Ours + Formulation 2 | -0.2066 | 1.1069 | 0.1138 | -0.1651 |
| Ours + Formulation 3 | -1.7045 | 0.246 | 1.0838 | 3.1921 |
| Ours + Formulation 1 + Interpolation | 0.9039 | 2.226 | -0.7365 | **-0.4403** |

## 5.2 Performance on a Smaller Dataset Setting

We examine the effects of kinematic priors on a smaller dataset size. This is motivated by the fact driving datasets naturally have an imbalance of scenarios, where many samples are representative of longitudinal straight-line driving or stationary movement, and much less are representative of extreme lateral movements such as U-turns. Thus, large and robust benchmarks like the Waymo, Nuscenes, and Argoverse datasets are necessary for learning good models.

However, large datasets are not always accessible depending on the setting. For example, traffic laws, road design, and natural dynamics vary by region. It would be infeasible to expect the same scale and robustness of data from every scenario in the world, and thus trajectory forecasting will run into settings with less data available.

We train the baseline model and all formulations on only 1% of the original Waymo dataset and benchmark their performance on 100% of the evaluation set in Table 2. All experiments were trained over 50 epochs. In the small dataset setting, we observe that providing a kinematic prior in any form improves performance for minimum final displacement error (minFDE). Additionally, we observe better performance across all metrics for formulations 1, 2, and 1 with interpolation. Overall, **formulation 1 provides the greatest boost in performance, with a nearly** $12\%$ **improvement in mAP,** $12.5\%$ **improvement in minADE,** $27.8\%$ **improvement in minFDE, and** $8.3\%$ **improvement in Miss Rate**.

Compared to the results from Table 1, the effects of kinematic priors in learning are much more pronounced. Since kinematic priors naturally relate the position at one timestep to the position at the next, we believe that the performance boost can be attributed to this natural relationship. In backpropagation, optimization of one position further into the time horizon directly influences predicted positions at earlier timesteps via the kinematic model. Without the kinematic prior, the relation between timesteps may be implicitly related through neural network parameters. When the model lacks data to form a good model of how an agent moves through space, the kinematic model can compensate to model such simple relationships.

Table 2: **% difference in performance on vehicles for each kinematic formulation versus SOTA in a small dataset setting.** Additionally, we run experiments to examine the effect of kinematic priors on settings with less data available. We train models on 1% of the original Waymo Motion Dataset and use the same full evaluation set as that in Table 1. From these results, we see more pronounced improvements in performance metrics in settings with significantly less data available, with a nearly **12% performance gain on the main benchmark metric (mAP) over the baseline**.

| Method | ($\Delta\%$) mAP$\uparrow$ | ($\Delta\%$) minADE$\downarrow$ | ($\Delta\%$) minFDE$\downarrow$ | ($\Delta\%$) MissRate$\downarrow$ |
|---|---|---|---|---|
| Baseline | 0 | 0 | 0 | 0 |
| Ours + Formulation 1 | **11.8444** | **-12.528** | -27.1767 | **-8.3266** |
| Ours + Formulation 2 | 6.7767 | -5.8432 | **-27.7645** | -7.2791 |
| Ours + Formulation 3 | -5.3035 | 30.6413 | -20.5494 | -0.8327 |
| Ours + Formulation 1 + Interpolation | 4.1839 | -7.3498 | -23.8817 | -4.808 |

## 5.3 Effect on Robustness

We also show how *kinematic priors can influence performance in the presence of noise*. This is inspired by the scenario where sensors may have a small degree of noise associated with measurements dependent on various factors, such as weather, quality, interference, etc.

We evaluate the models from Table 3 when input trajectories are perturbed by standard normal noise $n_\epsilon \sim \mathcal{N}(0, 1)$; results for performance degradation are shown in Table 3.

We compute the entries of Table 3 by measuring the % of degradation of the perturbed evaluation from the corresponding original clean evaluation. Interestingly, we find that Formulation 2 from Section 4.2.2 with acceleration components preserves the most performance. This may be due to that second-order terms like acceleration are less influenced by perturbations on position. Additionally, distributions of acceleration are typically centered around zero regardless of how position is distributed (Albaba & Yildiz, 2022).

Table 3: **Degradation of performance in the presence of noise.** We also compare the robustness of each method by measuring the impact on performance in the presence of noise. In the table, we compute the % change in performance between perturbed evaluation and clean evaluation, relative to each method. Additionally, we omit results for Formulation 1 with interpolation, as the noise added would not be proportional to the others for fair comparison. We observe that **Formulation 2, with acceleration components, offers the greatest relative boost in robustness over other formulations**.

| Method | $(\Delta\%)$ mAP↑ | $(\Delta\%)$ minADE↓ | $(\Delta\%)$ minFDE↓ | $(\Delta\%)$ MissRate↓ |
|---|---|---|---|---|
| Baseline | -4.9587 | 4.7965 | 1.6527 | 3.5223 |
| Ours + Formulation 1 | -4.9445 | 3.5542 | 1.4322 | 2.7072 |
| Ours + Formulation 2 | **-3.9596** | **2.4693** | **1.2800** | **2.7012** |
| Ours + Formulation 3 | -4.7294 | 3.3984 | 1.3447 | 2.5600 |

## 6 Discussion and Conclusion

In this paper, we present a simple and easy-to-implement method for including kinematic priors in probabilistic trajectory forecasting methods. Kinematic priors can also be trivially implemented for deterministic methods where linear approximations are not necessary. With no additional overhead, kinematic priors not only show improvement in models trained on robust datasets but also in suboptimal settings with small datasets and noisy trajectories, with up to 12% improvement in smaller datasets and 1% **less** performance degradation in the presence of noise for the full Waymo dataset. For overall performance improvement, we find Formulation 1 in Section 4.2.1 with velocity components to be the most beneficial and well-rounded to prediction performance.

We also observe that when there is large-scale data to learn a good model of how vehicles move, the effects of kinematic priors are less pronounced. This can be observed from the less obvious improvements over the baseline in Table 1 compared to Table 2. We conjecture that model complexity and dataset size will eventually out-scale the effects of the kinematic prior. With enough resources and high-quality data, trajectory forecasting models will learn to *"reinvent the steering wheel"*, or implicitly learn how vehicles move via the complexity of the neural network. For practical autonomous systems, portability and adaptability are important problems to consider aside into the future. We hope that kinematic priors can be considered in trajectory forecasting and simulation as deployment ventures past well-represented urban scenarios to suboptimal settings, whether it is with noisy sensors, smaller datasets to adapt from, or downscaled models for inference speed and portability.

For future work, we would like to further explore how kinematic priors can be used for transfer learning between domains, such as transfer from right-handed traffic to left-handed traffic. While distributions of trajectories may change in distribution and scale depending on the environment, kinematic parameters, especially on the second order, will remain more constant between domains. For example, while the magnitude of trajectory data may scale based on the speed of the vehicle and road scenario, acceleration may have nicer properties by being centered around zero and only varying in scale between scenarios.

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
