# A  APPENDIX

## A.1  ADDITIONAL FORMULATION: STEERING AND ACCELERATION

We also derive an additional second-order kinematic formulation: steering and acceleration. This formulation is the second-order version of the velocity-heading formulation. Here, we assume acceleration $a$ to be scalar and directionless, in contrast to Formulation 2, where we consider acceleration to be a vector with lateral and longitudinal components. Similarly to Formulation 2, we also use the lienar approximation of $\tan(\cdot)$ in order to derive approximated position distributions.

Following the Bicycle Model, the update for speed and heading at each timestep is:

$$\begin{bmatrix} s^{t+1} \\ \theta^{t+1} \end{bmatrix} = \begin{bmatrix} s^t + a \cdot \Delta t \\ \theta^t + \frac{s \cdot \tan(\delta)}{L} \cdot \Delta t \end{bmatrix}$$

Where $L$ is the length of the agent. When we represent this process probabilistically as random Gaussian variables, the formulation becomes:

$$\begin{aligned}
\mathcal{N}_{s^{t+1}} &= \left( \mu_s^t + \sigma_s^t \cdot \epsilon_s \right) + \left( \mu_a^t + \sigma_a^t \cdot \epsilon_a \right) \cdot \Delta t \\
\implies \mu_s^{t+1} &= \mu_s^t + \mu_a^t \cdot \Delta t, \\
\sigma_s^{t+1} &= \sigma_s^t + \sigma_a^t \cdot \Delta t \\
\mathcal{N}_{\theta^{t+1}} &= \left( \mu_\theta^t + \sigma_\theta^t \cdot \epsilon_\theta \right) + \frac{1}{L} \cdot T \left( \mu_\delta^t + \sigma_\delta^t \cdot \epsilon_\delta \right) \cdot \left( \mu_{s^t} + \sigma_s^t \cdot \epsilon_s \right) \cdot \Delta t \\
&= \left( \mu_\theta^t + \sigma_\theta^t \cdot \epsilon_\theta \right) + \frac{1}{L} \cdot \left( \tan(\mu_\delta^t) + \frac{1}{\cos^2(\mu_\delta^t)} \cdot (\sigma_\delta^t \cdot \epsilon_\delta) \right) \cdot \left( \mu_s^t + \sigma_s^t \cdot \epsilon_s \right) \cdot \Delta t \\
\implies \mu_\theta^{t+1} &= \mu_\theta^t + \frac{1}{L} \cdot \left( \mu_s^t \cdot \tan(\mu_\delta^t) \right) \cdot \Delta t, \\
\sigma_\theta^{t+1} &= \sigma_\theta^t + \frac{1}{L} \cdot \left( \mu_s^t \cdot \sigma_\delta^t \cdot \frac{1}{\cos^2(\mu_\delta^t)} + \sigma_s^t \cdot \tan(\mu_\delta^t) + \frac{1}{\sqrt{2}} \cdot \left( \sigma_s^t \cdot \sigma_\delta^t \cdot \frac{1}{\cos^2(\mu_\delta^t)} \right) \right) \cdot \Delta t
\end{aligned}$$

With the terms characterizing $\mathcal{N}_s^{t+1}$ and $\mathcal{N}_\theta^{t+1}$, this formulation then degenerates into Formulation 3 in Section 4.2.3.

## A.2 ADDITIONAL RESULTS BY CLASS

In the paper, we present results on vehicles since we use kinematic models based on vehicles as priors. Here, we present the full results per-class for each experiment in Tables 4, 5, 6, and 7. The results reported in the paper are starred (*), which are re-iterated below for full context.

Table 4: **Per-class results for performance on 100% of the Waymo Dataset.**

| Class | Method | (Δ%) mAP↑ | (Δ%) minADE↓ | (Δ%) minFDE↓ | (Δ%) MissRate↓ |
|---|---|---|---|---|---|
| Average | Baseline | 0 | 0 | 0 | 0 |
| | Ours + Formulation 1 | **1.7492** | -0.4455 | **-2.3882** | **-1.1098** |
| | Ours + Formulation 2 | -2.2235 | 0.1337 | -1.0881 | -0.7009 |
| | Ours + Formulation 3 | -0.9487 | **-0.4604** | -0.7560 | 1.2850 |
| | Ours + Formulation 1 + Interpolation | -0.5336 | 1.4553 | -1.6534 | 0.0000 |
| Vehicle* | Baseline | 0 | 0 | 0 | 0 |
| | Ours + Formulation 1 | **2.376** | **-0.3444** | **-0.9102** | -0.3853 |
| | Ours + Formulation 2 | -0.2066 | 1.1069 | 0.1138 | -0.1651 |
| | Ours + Formulation 3 | -1.7045 | 0.246 | 1.0838 | 3.1921 |
| | Ours + Formulation 1 + Interpolation | 0.9039 | 2.2260 | -0.7365 | **-0.4403** |
| Pedestrian | Baseline | 0 | **0** | 0 | 0 |
| | Ours + Formulation 1 | **0.4657** | 0.2343 | **-0.7773** | -2.7692 |
| | Ours + Formulation 2 | -1.2806 | 0.885 | -0.2065 | **-3.8974** |
| | Ours + Formulation 3 | 0.0873 | 0.9630 | 0.6072 | -1.9487 |
| | Ours + Formulation 1 + Interpolation | 0.3492 | 4.1385 | 1.4574 | 0.8205 |
| Cyclist | Baseline | 0 | 0 | 0 | 0 |
| | Ours + Formulation 1 | **2.4911** | -0.8991 | **-4.5646** | **-1.0656** |
| | Ours + Formulation 2 | -6.0854 | -1.1786 | -2.6418 | 0.1279 |
| | Ours + Formulation 3 | -1.2100 | **-1.8348** | -3.1439 | 1.0230 |
| | Ours + Formulation 1 + Interpolation | -3.5943 | -0.5832 | -3.9884 | 0.0000 |

Table 5: **Per-class results for performance on 1% of the Waymo Dataset.**

| Class | Method | (Δ%) mAP↑ | (Δ%) minADE↓ | (Δ%) minFDE↓ | (Δ%) MissRate↓ |
|---|---|---|---|---|---|
| Average | Baseline | 0 | 0 | 0 | 0 |
| | Ours + Formulation 1 | -1.6418 | **-6.3763** | **-14.9755** | **-5.173** |
| | Ours + Formulation 2 | **0.7463** | -0.7756 | -14.5275 | -2.9916 |
| | Ours + Formulation 3 | -6.1692 | 16.0354 | -11.2498 | -0.2493 |
| | Ours + Formulation 1 + Interpolation | -1.2438 | -2.1014 | -12.2530 | -0.0312 |
| Vehicle* | Baseline | 0 | 0 | 0 | 0 |
| | Ours + Formulation 1 | **11.8444** | **-12.5280** | -27.1767 | **-8.3266** |
| | Ours + Formulation 2 | 6.7767 | -5.8432 | **-27.7645** | -7.2791 |
| | Ours + Formulation 3 | -5.3035 | 30.6413 | -20.5494 | -0.8327 |
| | Ours + Formulation 1 + Interpolation | 4.1839 | -7.3498 | -23.8817 | -4.8080 |
| Pedestrian | Baseline | **0** | 0 | 0 | 0 |
| | Ours + Formulation 1 | -7.8373 | **-1.1325** | -1.2123 | 3.2820 |
| | Ours + Formulation 2 | -11.1275 | 1.7743 | -0.9999 | 6.2960 |
| | Ours + Formulation 3 | -3.2902 | -0.6795 | **-3.0440** | 0.1340 |
| | Ours + Formulation 1 + Interpolation | -8.0961 | 3.7750 | 1.3716 | 11.2525 |
| Cyclist | Baseline | 0 | 0 | 0 | 0 |
| | Ours + Formulation 1 | -5.5283 | **-1.6251** | **-4.6731** | **-5.3754** |
| | Ours + Formulation 2 | **14.2506** | 3.8549 | -2.8120 | -2.4949 |
| | Ours + Formulation 3 | -11.9165 | 6.4701 | -2.5166 | 0.1588 |
| | Ours + Formulation 1 + Interpolation | 4.4840 | 1.3908 | -2.6320 | 0.2041 |

Table 6: **Per-class performance degradation results with perturbed evaluation for models trained on 100% of the Waymo Dataset.**

| Class | Method | ($\Delta\%$) mAP↑ | ($\Delta\%$) minADE↓ | ($\Delta\%$) minFDE↓ | ($\Delta\%$) MissRate↓ |
|---|---|---|---|---|---|
| Average | Baseline | **-2.5793** | 2.5097 | 0.7066 | 1.6939 |
| | Ours + Formulation 1 | -2.9138 | 1.5662 | **0.3547** | **-0.6497** |
| | Ours + Formulation 2 | -3.2141 | **1.4385** | 0.9715 | 1.8824 |
| | Ours + Formulation 3 | -3.5917 | 1.7306 | 0.7760 | 0.6344 |
| Vehicle* | Baseline | -4.9587 | 4.7965 | 1.6527 | 3.5223 |
| | Ours + Formulation 1 | **-4.9445** | **3.5542** | **1.4322** | **2.7072** |
| | Ours + Formulation 2 | -3.9596 | 2.4693 | 1.2800 | 2.7012 |
| | Ours + Formulation 3 | -4.7294 | 3.3984 | 1.3447 | 2.5600 |
| Pedestrian | Baseline | **-1.1932** | 0.8329 | 0.0243 | 3.3846 |
| | Ours + Formulation 1 | -1.5933 | **-0.2077** | **-0.7344** | **-1.0549** |
| | Ours + Formulation 2 | -3.066 | 0.4902 | 0.1582 | 2.7748 |
| | Ours + Formulation 3 | -2.0646 | 0.4383 | 0.4587 | 2.7197 |
| Cyclist | Baseline | **-0.9253** | 1.0207 | 0.1255 | -0.5541 |
| | Ours + Formulation 1 | -1.6667 | **0.4537** | **-0.1614** | **-3.0590** |
| | Ours + Formulation 2 | -2.3494 | 0.8361 | 1.0608 | 0.9366 |
| | Ours + Formulation 3 | -3.8905 | 0.6560 | 0.3594 | -1.7300 |

Table 7: **Per-class performance degradation results with perturbed evaluation for models trained on 1% of the Waymo Dataset.**

| Class | Method | ($\Delta\%$) mAP↑ | ($\Delta\%$) minADE↓ | ($\Delta\%$) minFDE↓ | ($\Delta\%$) MissRate↓ |
|---|---|---|---|---|---|
| Average | Baseline | -3.4328 | 0.5411 | 0.1187 | **0.1246** |
| | Ours + Formulation 1 | -6.2721 | **0.1349** | **-0.1171** | 0.6901 |
| | Ours + Formulation 2 | **-3.3086** | 0.7635 | 0.3181 | 1.574 |
| | Ours + Formulation 3 | -4.0297 | 0.3420 | 0.8672 | 2.2805 |
| Vehicle | Baseline | -6.0695 | 1.1266 | **0.2207** | 0.9132 |
| | Ours + Formulation 1 | **-4.4784** | 1.0241 | 0.8180 | 1.2892 |
| | Ours + Formulation 2 | -5.6843 | 1.8236 | 1.3033 | 1.5933 |
| | Ours + Formulation 3 | -5.6005 | **-0.3117** | 0.4686 | **0.4605** |
| Pedestrian | Baseline | **-0.2957** | **-1.1136** | **-0.9734** | **-2.6122** |
| | Ours + Formulation 1 | -4.4124 | -0.3627 | -0.8599 | -0.0649 |
| | Ours + Formulation 2 | -3.0782 | -0.2411 | -0.3396 | 2.8355 |
| | Ours + Formulation 3 | -4.8930 | -0.3421 | -0.2008 | 2.9431 |
| Cyclist | Baseline | -5.8354 | 0.5518 | 0.4041 | **0.4083** |
| | Ours + Formulation 1 | -11.3134 | **-0.5455** | **-0.7410** | 0.5273 |
| | Ours + Formulation 2 | -1.3978 | 0.1019 | -0.3669 | 1.1398 |
| | Ours + Formulation 3 | **-0.6276** | 1.4908 | 1.6792 | 3.5779 |

## A.3 EXPERIMENT HYPERPARAMETERS

Table 8: **Model Architecture Hyperparameters**

| Component | Hyperparameter | Value |
|---|---|---|
| Encoder | # Hidden Features | 128 |
| | # Attention Layers | 2 |
| | # Attention Heads | 2 |
| | Local Attention | True |
| Decoder | Hidden Features | 128 |
| | # Decoder Layers | 2 |
| | # Attention Heads | 2 |
| | # Hidden Map Features | 64 |