# OpenReview forum: "Don't Reinvent the Steering Wheel"
_ICLR.cc/2024/Conference — Submitted to ICLR 2024_

### Official Review · Reviewer_NxgX · 2023-10-16

**Soundness:** 1 poor
**Presentation:** 2 fair
**Contribution:** 1 poor
**Rating:** 3
**Confidence:** 5

**Summary:**

This paper explores several alternative output representations of the trajectory prediction model. Instead of directly predicting the position coordinates (x, y), the authors proposed three alternative output representations:
* Predicting (vx, vy)
* Predicting (ax, ay)
* Predicting (speed, heading)

The authors implemented the three approaches on the Motion Transformer (MTR) model. The evaluation result on the WOD dataset shows that by predicting (vx, vy), they improve the prediction mAP by 2.376%.

**Strengths:**

* This paper shows some performance improvement over the MTR model on the WOD dataset.

**Weaknesses:**

* The contribution of this paper is very weak. The related work section claims that none of the existing trajectory works uses the bicycle kinematic model, which is not true. There are many existing trajectory prediction models (such as MultiPath++ [1] and DKM [2]) that explicitly predict future accelerations and steering angles and roll out future trajectories using the bicycle kinematic model.

[1] Varadarajan et al. 2022. Multipath++: Efficient information fusion and trajectory aggregation for behavior prediction.

[2] Cui et al. 2019. Deep Kinematic Models for Kinematically Feasible Vehicle Trajectory Predictions.

* This paper talks about the advantages of using the bicycle kinematic model as the prior for trajectory prediction. However, none of the three proposed formulations used the actual bicycle kinematic model. From the result in Table 1, Formulation 1 has the best performance, and Formulation 1 is simply to predict deltas between waypoints.

* The evaluation result lacks some important information. The tables only show the relative improvement from the MTR baseline. However, it is not clear whether the MTR numbers are from the MTR paper, the WOD leaderboard, or from the authors' own implementation. I highly recommend the authors submit their solution to the WOD leaderboard and report the numbers from the leaderboard.


--- Other minor comments

* Please add equation numbers.

* I am not sure sigma_{x + vx * dt} = sigma_x + sigma_{vx} * dt is correct. The summation of two Gaussian variables x + y should be sigma_{x + y} = \sqrt{sigma_x ^ 2 + sigma_y ^ 2}.

* Reference Chai et al., 2020a and Chai et al., 2020b are the same paper.

**Questions:**

N/A

---

> ### Author Response · Authors · 2023-11-20
>
> Thank you for your feedback! We will make sure to add revisions for your suggestions accordingly, especially in the related works section. Additionally, please see the general response for context of our work w.r.t. previous methods. We will make sure to include both writing and result comparisons in future revisions. We address specific questions below:
>
> - “The tables only show the relative improvement from the MTR baseline. However, it is not clear whether the MTR numbers are from the MTR paper, the WOD leaderboard, or from the authors’ own implementation.”
>
> The baseline MTR numbers represent our own re-training of a downscaled MTR model, whose absolute performance results will be included in future revision. Here are the absolute baseline numbers for your convenience:
>
> Baseline (Downscaled and re-trained MTR)
>
> mAP - 0.3373
>
> minADE - 0.6734
>
> minFDE - 1.4153
>
> MissRate - 0.1712
>
> Original MTR (Full-size, not re-trained)
>
> mAP - 0.4129
>
> minADE - 0.6050
>
> minFDE - 1.2207
>
> MissRate - 0.1351
>
> - “None of the three proposed formulations used the actual bicycle kinematic model.”
>
> We performed derivations and experiments in increasing order of complexity (first-order formulations involving velocity and heading first, then second-order formulations). We will include further results centered around steering and acceleration in future revisions.
>
> - “I am not sure sigma_{x + vx * dt} = sigma_x + sigma_{vx} * dt is correct. The summation of two Gaussian variables x + y should be sigma_{x + y} = \sqrt{sigma_x ^ 2 + sigma_y ^ 2}.”
>
> Thank you for catching this! We will fix this and update the paper.

---

> > ### Comment · Reviewer_NxgX · 2023-11-20
> > **Thank you for your response**
> >
> > Thank you for your response. I will keep my score.

---

### Official Review · Reviewer_i6he · 2023-10-28

**Soundness:** 3 good
**Presentation:** 2 fair
**Contribution:** 1 poor
**Rating:** 3
**Confidence:** 4

**Summary:**

The authors develop a trajectory forecasting model that makes use of a simple kinematics model for vehicle dynamics to induce a bias and facilitate learning. In particular, instead of predicting a GMM over x/y, they predict 1) the velocity, 2) acceleration, or 3) speed and heading. For 3) they also derive analytical error bounds for a linear approximation around sin / cos. They show that formulation 1) improves a baseline on WOMD.

**Strengths:**

* The proposed model beats the baseline model
* Results for different kinematics models

**Weaknesses:**

* The authors claim that previous SotA results were only obtained by directly predicting Euclidean coordinates. However, for example “MULTIPATH++: EFFICIENT INFORMATION FUSION AND TRAJECTORY AGGREGATION FOR BEHAVIOR PREDICTION” (https://arxiv.org/pdf/2111.14973.pdf), which the authors cite as a SotA model, also uses an underlying kinematics model as described in Section 3.6.
* Lack of novelty / contributions. Trajectory forecasting with kinematics models exists in literature. The given kinematic model derivations are quite straightforward.

**Questions:**

As mentioned above, kinematics models have been applied for ML trajectory forecasting and hence, the contributions of this paper seem minor.

---

> ### Author Response · Authors · 2023-11-20
>
> Thank you for your suggestions. We clarify how we approach kinematic priors in our method differently from existing work in the general response above, and will make sure to incorporate revisions accordingly in the paper. We agree that the idea of using kinematic priors is not new to trajectory forecasting tasks, and thus will re-frame the contributions accordingly, as we believe previous work considered kinematic models *deterministically* rather than *probabilistically*, which remains as our key contribution over prior works and the key differentiator.  This formulation offers several benefits as we stated above in the general response.
>
> While the kinematic model derivations are straightforward, the additional novelty is the consideration of the relationship between variances of kinematic variables and resulting trajectories. Mean kinematic variables and mean trajectories are considered in previous work directly, but the *disadvantage is the loss of expressiveness reflected in trajectory distributions when assuming a Gaussian model of decision making*.   *This issue* is what we address in the paper.
>
> Hopefully this answers some of your questions, and thank you for your review!

---

### Official Review · Reviewer_uGDE · 2023-10-29

**Soundness:** 2 fair
**Presentation:** 2 fair
**Contribution:** 2 fair
**Rating:** 3
**Confidence:** 4

**Summary:**

This paper proposes a method for using kinematic models for trajectory forecasting in the context of self-driving. Three different kinematic formulations (velocity, acceleration, speed+heading) are ablated in three settings (full dataset, small dataset, noise) on the Waymo open motion forecasting dataset. Analytical error bounds some of the kinematic formulations given.

**Strengths:**

* The method proposed makes sense and is simple.
* The analysis on dataset size and noise was nice to see, and not common in existing literature to my knowledge.
* The derivations for the linear approximations for the distribution of positions is also useful.

**Weaknesses:**

**1) Related work**

This paper claims the novelty for "simple and effective method for incorporating kinematic priors into probabilistic models
for trajectory forecasting", but is missing many related works that have explored this in the past. For some examples of works which use some variant of bicycle, unicycle or other kinematic model for trajectory forecasting in the context of self driving:

* Imagining The Road Ahead: Multi-Agent Trajectory Prediction via Differentiable Simulation
* Deep Kinematic Models for Kinematically Feasible Vehicle Trajectory Predictions
* A Kinematic Model for Trajectory Prediction in General Highway Scenarios
* MixSim: A Hierarchical Framework for Mixed Reality Traffic Simulation
* Guided Conditional Diffusion for Controllable Traffic Simulation

As these are papers that simply come top of mind / found after a brief literature review, I may have missed more and thus I encourage the authors to add this list after performing a more thorough literature review. For the rebuttal I'd like to see the authors more clearly position their work in the context of existing literature.


**2) Experiments**
The paper was unclear about the baseline performance. Specifically, did the authors compare to the original MTR, or a re-implemented and retrained version of MTR? I found the following statements in the paper:
> We implement kinematic priors on state-of-the-art method Motion Transformer (MTR) Shi et al. (2022), which serves as our baseline method.

> ... we downscale the model from its original size of 65 million
parameters to 2 million parameters and reproduce baseline results in our experiments

> We compare our model against the state-of-the-art architecture...

In all tables in both the main paper and the appendix, the results are written as relative to the baseline, and I could not find absolute numbers for the baseline anywhere. Thus I am lead to believe that the results are for the smaller model, which likely performs worse than the original MTR? If this is true, I believe the authors need to release the absolute performance of their baseline.
The reason that the absolute performance is important to me is that its unclear whether the gains of the kinematic model hold when absolute performance increases as you scale the model, etc.

**Questions:**

Could the authors address my questions in the Weaknesses section.

---

> ### Author Response · Authors · 2023-11-20
>
> We thank you for the feedback, as well as the additional references. We contextualize our work with the provided works in the general response and answer specific questions here.
>
> - “Specifically, did the authors compare to the original MTR, or a re-implemented and retrained version of MTR?”
>
> We compare it to the re-implemented and re-trained version of MTR, which can be expected to have lower performance due to the sheer difference in number of parameters. We do this mostly due to limitations on available computational resources.
>
> - “I could not find absolute numbers for the baseline anywhere”.
>
> We should have included this in the appendix and will add it in revisions. For your convenience, we include the absolute performance of the downscaled baseline here.
>
> Baseline (Downscaled and re-trained MTR)
>
> *mAP* - 0.3373
>
> *minADE* - 0.6734
>
> *minFDE* - 1.4153
>
> *MissRate* - 0.1712
>
> Original MTR (Full-size, not re-trained)
>
> *mAP* - 0.4129
>
> *minADE* - 0.6050
>
> *minFDE* - 1.2207
>
> *MissRate* - 0.1351
>
> - “The reason that the absolute performance is important to me is that its unclear whether the gains of the kinematic model hold when absolute performance increases as you scale the model, etc.”
>
> We agree with this intuition, and also observe from preliminary training curves that gains from the kinematic model diminish as the model size increases. However, we believe that state-of-the-art models should also consider models with smaller numbers of trainable parameters for the sake of *practicality and portability*. While the benefit of kinematic priors diminish with greater model sizes, we believe that they will be beneficial in the event that such models need to be downscaled and fit onto a robot.
>
> We hope this answers some of your questions, and thank you again for the feedback!

---

> > ### Comment · Reviewer_uGDE · 2023-11-21
> > **Thank you for your response**
> >
> > Thank you for addressing my comments. Given that a new version which includes all the feedback would be significantly different and may involve additional experimental results, analysis, etc. I believe I should stay with my original recommendation and encourage the authors to resubmit in the future.

---

### Official Review · Reviewer_AmF6 · 2023-11-01

**Soundness:** 1 poor
**Presentation:** 2 fair
**Contribution:** 1 poor
**Rating:** 1
**Confidence:** 5

**Summary:**

The authors present a method to improve trajectory forecasting using kinematic models. In particular, they propose to slightly change the output of the existing models where they implement kinematic equations to ensure that feasible trajectories are inferred. They discuss several different kinematic approaches and showcase how this idea can be applied to a recent state-of-the-art model.

**Strengths:**

- A very relevant problem being investigated.
- Intuitive idea being proposed.
- Promising experimental results shown.

**Weaknesses:**

- Novelty is very limited, as the authors present a method of an existing work that is not referenced.
- Writing and explanations can be improved.
- Experiments are quite dry and could be expanded with more visualizations.

**Questions:**

The work presents quite an intuitive idea that is showing good results. However, the problem is that the same idea is already presented in an earlier work: "Deep Kinematic Models for Kinematically Feasible Vehicle Trajectory Predictions", ICRA 2020. The authors do not cite this paper and do not put their work in the correct context when it comes to the existing literature. As such, the main contribution of the work, as stated by the authors, lacks novelty.
Please find detailed comments below:
- "but also scales well in size", unclear if this is a good or a bad thing. The authors should clarify this better.
- The authors need to cite the earlier work listed above, and put their work into context. This is the biggest issue with the current manuscript and makes the work not ready for publication until this is done (since the level of contributions of the work is questionable).
- Figures 1 and 2 are not referenced in the text.
- Notation can be improved. E.g., in the equation for L  in Section 4.1 the notation p_h is not introduced before being used. The authors should make sure that all notation is properly defined.
- "we downscale the model", how exactly? Unclear.
- The authors do not properly explain the experiments, such as the number of mixture components used and other information relevant to understanding the approach and the experiments.
- For the experiments with noisy trajectories, do you add noise during training as well? Unclear, would be good to clarify.
- The experiments are somewhat dry, would be good to add some visual results as well.

---

> ### Author Response · Authors · 2023-11-20
>
> We thank you for the detailed and thoughtful review! We explain the difference between previous work and ours in the general response, but will use this response to clarify specific questions. Writing updates will be directly revised in the paper, and we will be sure to add visual results as well.
>
> - “but also scales well in size”; is this a good thing or bad thing?
>
> We mention this as a good thing in the context of performance, but also a bad thing in the context of portable autonomous driving models for real-time inference. We observe trends in recent years that SOTA models tend to become larger and larger–combining SOTA from several tasks together for full-stack autonomous driving may prove to be computationally costly if model sizes continue to scale along with performance. We would like to motivate work in the direction of smaller models with more hand crafted kinematic modeling or classical simulation techniques for sake of *portability* or unseen domain situations with data scarcity (while still preserving performance).
>
> We will clarify this confusion in future revisions of the paper.
>
>
> - “we downscale the model” how?
>
> Since MTR is a transformer-based model, we downscale the model directly by reducing the number of heads, number of encoder and decoder layers, and hidden features. The full details on model parameters used for results can be found in Table 8 of the appendix. We will reference this Table in the revision.
>
>
> - “For the experiments with noisy trajectories, do you add noise during training as well?”
>
> For experiments regarding noisy trajectories, we do not add noise during training. The experiments were inspired by the scenario where training data is clean and well-processed, but sensors may be noisy in the real world. This experiment examined the robustness of our model in comparison to direct prediction on positional waypoint trajectories.
>
> We hope these answers help clarify your questions, and again, thank you for the helpful and detailed feedback!

---

> > ### Comment · Reviewer_AmF6 · 2023-11-21
> >
> > Thank you for addressing my comments.
> > Looking at the other reviews and the responses, a new version of the paper that would incorporate all the feedback would require a full re-review before acceptance. As such, I would like to stay with my original recommendation and encourage the authors to address the many comments and resubmit to a future venue.

---

### Author Response · Authors · 2023-11-20

We thank all the reviewers for their time and suggestions in the detailed and helpful reviews. We agree that the paper can benefit from more comparisons and context to current work to distinguish itself in terms of contribution and novelty and we will revise the paper accordingly. Here, we address common concerns and clarifications across multiple reviews:

1) ”The paper lacks references and context to previous work.”

Reviewers provided references to work not cited in our submission.  The lack of discussion and results against these works is an oversight in the current version; we'll add suggested references and comparison to better contextualize quantitative performance.

However, we would also like to explain the key differences between our work and the missing prior works. Most notably, our work shares close resemblance with that of Cui et al., in "Deep Kinematic Models for Kinematically Feasible Vehicle Trajectory Predictions''. Cui et al. introduce a recurrent kinematic layer which unrolls vehicle trajectories from kinematic variable predictions via time integration.

Although this idea at first appears to be very similar to our work, the key difference between this paper and ours, however, is that ours model predict the unrolling of Euclidean trajectories *probabilistically* rather than deterministically. This key difference is also common throughout other referenced works, even if the model architectures involve probabilistic predictions (Multipath++, Imagining the Road Ahead, MixSim).

Previous works use kinematic models as a direct transformation on either a single trajectory prediction or the mean in a distribution of trajectories, which neglect the transformation on the spread, or formally “variance”, of the resulting Euclidean position trajectories. The standard deviation or correlation coefficient is often treated as a hand-tuned hyperparameter or a learnable parameter, such as in the case of Multipath++. In the case of “Guided Conditional Diffusion for Controllable Traffic Simulation”, new traffic simulation behaviors are generated in a diffusion-based method by denoising states defined in terms of kinematic variables. We argue that, in this case, the task and approach is largely different from ours, and that the use of kinematic variables here is not trivially applicable to model probabilistic trajectory prediction (which is impractical via diffusion). But, more importantly, our approach takes into account of probabilistic variability of trajectory prediction (due to individual vehicle states, kinematics, sensing, driver behavior, etc), thereby increasing the probability distribution function of data coverage, while diffusion-based methods are known to not change the the probability distribution function of coverage. This is a significant and important difference with implication in robustness of learning, prediction, and control of autonomous vehicles.

2) “Kinematic models have already been used in previous work.”

Our wording in the related work is an oversight, and we agree it should be clarified in future revisions.
Although kinematics in trajectory prediction has been proposed in several works, in short previous methods (including Cui et al. 2020) do not consider the transformation of variance from predicted Gaussian distributions of kinematic variables to distributions of vehicle positions. Time integration is not a linear transformation and thus, transforming a Gaussian of kinematic variables does not produce a Gaussian of vehicle trajectories. While it’s possible to relegate the handling of covariance to gradient descent or the human training the network, we derive an analytical linearly approximated relationship. This is one of our key contributions.

Why should we model kinematic model inputs as Gaussian distributions rather than deterministic predictions? Intuitively, we believe the inputs to a kinematic model may be better represented as a Gaussian distribution rather than as a deterministic value. The resulting trajectories unrolled from distributions of kinematic variables should be much more expressive than using the mean only.
Since time integration is a non-linear transformation, this unrolling of predicted Gaussians for kinematic variables does not produce Gaussian distributions of positional trajectories. For the sake of complementing existing architectures and SOTA methods, we compute the linear approximation of such transformations, in order to get an approximately accurate \sigma and preserve expressiveness of future trajectories as much as possible.

We agree with reviewers in that this should be more clearly delineated in the paper, both in writing and in results. We will ensure to include further comparison against these methods with additional results, across axes of trajectory prediction expressiveness and performance. Additionally, we will include results with steering and acceleration formulations which directly correspond to the kinematic bicycle model.

---

### Meta-Review · Area_Chair_ubZQ · 2023-12-06

**Metareview:**

Summary of paper: This paper investigates the use of kinematic vehicle dynamics as a way to structure deep-learning based predictors. It compares various models (e.g., velocity control vs. acceleration control) against the Motion Transformer (MTR) model.

Strengths: All reviewers agree that kino-dynamically feasible trajectory prediction is important, and appreciate the comparison to the MTR predictor.

Weaknesses: All reviewers raise the same concern that the core technical idea—of using kinodynamic vehicle models to structure prediction—is widely used to date, making the core contributions unclear. Additionally, all reviewers agree that more baselines that use vehicle models should be benchmarked against. I suggest evaluating against [1], which is an open-source implementation of a predictor which predicts actions and then integrates the vehicle dynamics to get feasible trajectory predictions.

[1] Salzmann, Tim, et al. "Trajectron++: Dynamically-feasible trajectory forecasting with heterogeneous data." Computer Vision–ECCV 2020: 16th European Conference, Glasgow, UK, August 23–28, 2020, Proceedings, Part XVIII 16. Springer International Publishing, 2020.

**Justification For Why Not Higher Score:**

After I carefully reviewed the discussion between the authors and reviewers, as well as the manuscript, I agree with the reviewer's recommendation for rejection at this time---the proposed approach is not substantially different from what is already done in off-the-shelf prediction models. With more comparisons to stronger baselines (e.g., methods that use vehicle models during prediction) and a clearer insight onto why the proposed incorporation of vehicle models is different, this paper could be strengthened for a future venue.

**Justification For Why Not Lower Score:**

N/A

---

### Decision · Program_Chairs · 2024-01-16

Reject